# Quantification of Tumor Hypoxia through Unsupervised Modelling of Consumption and Supply Hypoxia MR Imaging in Breast Cancer

**DOI:** 10.3390/cancers14051326

**Published:** 2022-03-04

**Authors:** Torgeir Mo, Siri Helene Bertelsen Brandal, Alvaro Köhn-Luque, Olav Engebraaten, Vessela N. Kristensen, Thomas Fleischer, Tord Hompland, Therese Seierstad

**Affiliations:** 1Faculty of Clinical Medicine, University of Oslo, 0316 Oslo, Norway; sibert@ous-hf.no (S.H.B.B.); olav.engebraten@medisin.uio.no (O.E.); v.n.kristensen@medisin.uio.no (V.N.K.); 2Department of Cancer Genetics, Institute for Cancer Research, Oslo University Hospital, 4950 Oslo, Norway; thomas.fleischer@rr-research.no; 3Department of Breast Diagnostic, Oslo University Hospital, 0379 Oslo, Norway; 4Oslo Centre for Biostatistics and Epidemiology, Faculty of Medicine, University of Oslo, 0316 Oslo, Norway; a.k.luque@medisin.uio.no; 5Department of Oncology, Oslo University Hospital, 0379 Oslo, Norway; 6Department of Tumor Biology, Institute for Cancer Research, Oslo University Hospital, 4950 Oslo, Norway; 7Department of Medical Genetics, Oslo University Hospital, 0450 Oslo, Norway; 8Department of Radiation Biology, Norwegian Radium Hospital, Oslo University Hospital, 4950 Oslo, Norway; tord.hompland@rr-research.no; 9Department of Research and Development, Division for Radiology and Nuclear Medicine, Oslo University Hospital, 0379 Oslo, Norway; therese.seierstad@medisin.uio.no

**Keywords:** hypoxia, magnetic resonance imaging, diffusion weighted imaging, CSH

## Abstract

**Simple Summary:**

Hypoxia in solid tumors is common in most solid cancers and is associated with treatment resistance to both chemo- and radiation-therapy. There is also reason to believe that hypoxia is an important determinant of metastic disease. Identifying hypoxia in solid tumors is important in treatment planning and decision making. In 2018 Hompland et al. proposed a method, based on quantifying consumption and supply of oxygen from diffusion weighted magnetic resonance imaging, to estimate the hypoxic fraction of a solid tumor. The method was based on training model parameters on a known hypoxia state in prostate cancer. In the present study we verified the validity of the consumption and supply concept in breast cancer. Furthermore, we developed and validated a new approach to the concept that does not require a ground truth to train the parameters.

**Abstract:**

The purpose of the present study is to investigate if consumption and supply hypoxia (CSH) MR-imaging can depict breast cancer hypoxia, using the CSH-method initially developed for prostate cancer. Furthermore, to develop a generalized pan-cancer application of the CSH-method that doesn’t require a hypoxia reference standard for training the CSH-parameters. In a cohort of 69 breast cancer patients, we generated, based on the principles of intravoxel incoherent motion modelling, images reflecting cellular density (apparent diffusion coefficient; ADC) and vascular density (perfusion fraction; fp). Combinations of the information in these images were compared to a molecular hypoxia score made from gene expression data, aiming to identify a way to apply the CSH-methodology in breast cancer. Attempts to adapt previously proposed models for prostate cancer included direct transfers and model parameter rescaling. A novel approach, based on rescaling ADC and fp data to give more nuanced response in the relevant physiologic range, was also introduced. The new CSH-method was validated in a prostate cancer cohort with known hypoxia status. The proposed CSH-method gave estimates of hypoxia that was strongly correlated to the molecular hypoxia score in breast cancer, and hypoxia as measured in pathology slices stained with pimonidazole in prostate cancer. The generalized approach to CSH-imaging depicted hypoxia in both breast and prostate cancers and requires no model training. It is easy to implement using readily available technology and encourages further investigation of CSH-imaging in other cancer entities and in other settings, with the goal being to overcome hypoxia-induced resistance to treatment.

## 1. Introduction

Breast cancer is the most commonly diagnosed cancer in women worldwide (International Agency for Research on Cancer (IARC) December 2020). The treatment options upon initial diagnosis are determined based on receptor and molecular status, local tumor extent, involvement of lymph nodes and tumor histology and grading. Despite all of these assessments it is not possible to identify who will relapse and whose tumor will metastasize [1]. This is an important clinical challenge because about 90% of breast cancer deaths are due to metastases, reflecting insufficient response to currently available treatments. There is increasing evidence that tumor hypoxia, areas of low oxygen concentration, is an important determinant of metastases [2] and an adverse indicator for patient prognosis independent of available prognostic parameters in use today [3].

Several methods have been used to measure tumor hypoxia in research settings [4], but due to inherent limitations none of these have been established as a clinical routine assay. Direct needle electrode measurements are invasive, limited to accessible tumors only, and cannot differentiate pO2 values from necrotic or viable hypoxic cells [5]. Furthermore, similarly to biopsy-based molecular biomarkers, the technique is highly operator-dependent and prone to sampling bias [6]. Positron emission tomography (PET) imaging of nitromidazole (FMISO) uptake depicts the intratumor heterogeneity of hypoxia but has limited spatial resolution and contrast, requires radiation exposure, long wait before scanning due to slow clearance from the blood yielding and high physiologic uptake in liver, intestine and kidney that prohibits abdominal imaging. As opposed to PET, magnetic resonance imaging (MRI) is readily available, non-invasive, does not rely on the use of ionising radiation, has high spatial resolution and is currently included in the work-up of most solid cancers, including breast cancer. Thus, a MRI-based method that depicts the heterogeneity of hypoxia in space and time will pave the way for development of new and innovative hypoxia modification therapies that may ultimately translate into improved treatment outcome.

In 2018, Hompland and colleagues presented a novel MR-based imaging strategy for depicting tumor hypoxia in a cohort of prostate cancer patients—Consumption and Supply Hypoxia (CSH) imaging [7]. CSH imaging exploits the intravoxel incoherent motion (IVIM) separation of signal loss from vascular pseudo diffusion and extracellular diffusion from diffusion weighted (DW) MR images to model hypoxia based on a combination of oxygen consumption (cell density) and oxygen supply (blood vessel density). Hypoxic fractions were calculated as the fraction of pixels within each tumor with high consumption and low supply. These hypoxic fractions were strongly correlated to tumor hypoxic fractions, assessed by pimonidazole staining of the resected prostate glands, and tumor aggressiveness. In a study of 74 patients with cervical cancer Hillestad et al. found that CSH imaging could predict chemoradiotherapy outcome [8]. Both studies used pimonidazole staining of tissue specimens as a direct measure of hypoxia, as reference standard for model-training [7,8]. The need for a reference standard is a major limitation for widespread CSH imaging of hypoxia and to what extent the CSH parameters found for prostate cancer are transferable to other cancer entities without re-training needs to be established.

Breast and prostate cancers have many similarities. They both arise in glandular tissues, with comparable morphology, and they display similar biology [9]. In this study we investigate if CSH-imaging can depict breast cancer hypoxia, using the CSH-method initially developed for prostate cancer. Furthermore, we propose a more generalized pan-cancer application of the CSH-method that does not rely on a reference standard of hypoxia for training the CSH-parameters.

## 2. Materials and Methods

### 2.1. Study Cohort

The study cohort consisted of 69 breast cancer patients recruited at Oslo university hospital as part of a multicentre study between November 2008 and July 2012 (ClinicalTrials ID NCT00773695) and included large (≥2.5 cm) untreated HER2-negative mammary tumors. Patient and tumor and characteristics are shown in Table 1.

Written informed consents were obtained from all patients prior to inclusion. The study protocol was approved by the institutional protocol review board, the regional ethics committee, the Norwegian Medicines Agency and carried out in accordance with the Declaration of Helsinki, International Conference on Harmony/Good Clinical practice.

### 2.2. Molecular Hypoxia Reference Standard

For each patient, a molecular hypoxia reference standard was established by calculating a molecular hypoxia score ($HS_{mol}$) from the mean expression of 15 genes identified by Buffa et al. [10]. mRNA profiling was performed in this study cohort by Silwal-Pandit et al. [11]. Right after the MRI examination was done, three core needle biopsies were collected from each patient. The biopsies weighed between 6–30 mg and were prioritized for RNA and DNA extraction. Gene expression profiling was performed using 40 ng total RNA and one color SurePrint G3 Human GE 8 × 60 k Microarrays (Agilent Technologies) following the manufacturer’s protocol. Feature values were log2-transformed, and replicated probes averaged to yield one value per probe. The data was quantile normalized, and missing values were imputed using a linear least squares regression with a cluster size k = 20. Batch, hospital (N = 3), RNA integrity number, and array background signal effects were removed using a generalized linear model to adjust the signal. The data were then mean centered in both patient-, and gene axis, on a cohort consisting of 131 patients, of which 69 are included in this study. Microarray data are previously published [11], and are available in the ArrayExpress database (http://www.ebi.ac.uk/arrayexpress, (accessed on 17 January 2022)) under accession number E-MTAB-4439. Based on $HS_{mol}$ patients were stratified into two groups. Because there is no consensus on the definition of a hypoxic tumor neither based on oxygenation level nor hypoxic tumor fraction [12], we chose to use a median split to divide the tumors into two equal groups referred to as less hypoxic and more hypoxic, similarly to the approach used by previous $pO_{2}$ studies [13].

### 2.3. MRI Examination

The MRI images were acquired with an ESPREE 1.5T MR scanner (Siemens, Erlangen, Germany) equipped with a phased-array bilateral breast coil (CP breast coil, Siemens, Erlangen, Germany). The MRI protocol follows the EUSOBI recommendations [14] and contained T2-weighted, diffusion-weighted, and dynamic contrast-enhanced (DCE) MRI. The DW MR images, forming the basis for constructing the CSH-images, were acquired using a single-shot spin-echo echo planar imaging (SE EPI) sequence with fat-saturated, short T1 inversion recovery (TR = 5200 ms, TE = 69 ms, field of view (FoV) = 360 mm × 185, pixel size = 2.57 mm × 2.57 mm, slice thickness = 4.5 mm) and diffusion gradients applied along three orthogonal directions with five *b*-values of 0, 50, 250, 500, and 800 s/mm${}^{2}$. DCE-MRI, used for guiding tumor delineation, was acquired using a k-space weighted spoiled gradient echo (TR = 5.46 ms, TE = 2.59 ms), with spectral adiabatic inversion recovery (SPAIR) fat suppression and spatial resolution 1 mm × 1 mm × 1.5 mm, following a bolus injection of the contrast agent Gadovist (0.08 mmol/kg).

### 2.4. Image Analysis

An overview of the steps involved in the analyses of the breast cancer images is shown in Figure 1.

First, the breast tumors were delineated on a noise reduced version of the b800 image. Image noise was reduced using an edge preserving anisotropic diffusion algorithm, and delineation was done using a marching-squares algorithm with an iso-level set by a local Otsu-threshold [15] in the region surrounding and including the tumor. An experienced breast radiologist (SHBB) manually reviewed the delineations using DW and DCE MR images as guidance and made corrections if needed. Next, the voxels within the resulting 3D volume were subjected to voxel-wise IVIM-modelling separating signal attenuation due to microcirculation-induced spin dephasing from thermal diffusion-induced spin dephasing [16]. The model can be described by a bi-exponential equation: $\frac{S}{S_{0}}=f_{p}e^{-bD^{*}}+\left(1-f_{p}\right)e^{-bADC}$ where $\frac{S}{S_{0}}$ is the signal normalized to the signal at zero diffusion-weighting, $f_{p}$ is the perfusion fraction, in the tissue of interest, $D^{*}$ is the pseudo-diffusion coefficient associated to the IVIM-effect and ADC is the apparent diffusion coefficient.

By assuming the signal lost due to the fast flow in the vasculature is large compared to the signal lost by Brownian motion in the extravascular space ($D^{*}\gg ADC$), the IVIM model can be simplified to a mono-exponential equation for high *b*-values: (1)$$\frac{S}{S_{0}}=\left(1-f_{p}\right)e^{-bADC}$$
for b≥200 s/mm${}^{2}$. DW images have inherently low signal-to-noise ratio (SNR), and with only one data point in the range dominated by flow-driven signal loss (b50), Estimates of $D^{*}$ and $f_{p}$ tends to be unstable when using the full IVIM model. More stable estimates of ADC and $f_{p}$ are obtained by using the simplified IVIM model described by Equation (Equation 1), including only *b*-values b≥200. Since $D^{*}$ is not required for the CSH model, we calculated ADC and $f_{p}$ by a voxel-by-voxel linear least squares fit to the MR signal at b>200 s/mm${}^{2}$. Voxels with an $ADC>2\times 10^{-3}$ mm${}^{2}$/s were considered necrotic [17], and excluded from further analysis.

Images of ADC and $f_{p}$ were then combined to generate estimates of the tumor hypoxic fractions. Hypoxic fractions were defined as the fraction of voxels that satisfies Equation (Equation 2). This is equivalent to the fraction of voxels that falls on the lower left side of the decision boundary in a plot of $f_{p}$ versus ADC, defined uniquely by the model coefficients $ADC_{0}$ and $f_{p,0}$ from Hompland et al. [7]:(2)$$\frac{ADC}{ADC_{0}}+\frac{f_{p}}{f_{p,0}}<1$$

The relative weights of ADC and $f_{p}$ on the resulting estimates of hypoxic fractions are determined by the model coefficients $ADC_{0}$ and $f_{p,0}$. Increase in either $ADC_{0}$ or $f_{p,0}$ would result in decreased relative weights of ADC and $f_{p}$ respectively in the final hypoxic fraction.

There is reason to believe that inherent differences in structural and perfusion properties between healthy prostate and breast tissue would hinder the direct application of a CSH model, developed and optimized in prostate, in breast tissue. A failure of such a model to identify hypoxia in breast cancer without alterations is therefore to be expected. Two different strategies were employed to overcome the challenge of different inherent tissue structure: The first strategy involved re-scaling the model coefficients to the new breast tissue specific distributions of the IVIM parameters. Defining new, tissue specific, model parameters in accordance with Equation (Equation 3) ensures a decision boundary that gives a relative weighting of consumption and supply that more closely resembles that found to be optimal in prostate.
(3)$$\begin{array}{c} ADC_{0}^{breast}=\hat{ADC}{\left(\frac{ADC_{0}}{\hat{ADC}}\right)}_{prostate}\\ f_{p,0}^{breast}={\hat{f}}_{p}{\left(\frac{f_{p,0}}{{\hat{f}}_{p}}\right)}_{prostate} \end{array}$$
where $\hat{ADC}$ and ${\hat{f}}_{p}$ are the mean values of all tumor voxels in all patients in each cohort.

Model parameter re-scaling is essentially equivalent to re-scaling the data itself, using a mean value normalization. This is a common method of dealing with difference in data dimensionality, but when comparing different cancer types, it is ignoring the known difference in oxygenation between the two tissues [18]. Different levels of oxygenation in different cancers have shown to be predictive of clinical outcome. Additionally, different cancers display different biological responses to similar oxygenation levels. Therefore, defining a tumor as hypoxic is not trivial. A breast cancer tumor can be categorized as hypoxic at an oxygenation level that would be considered non-hypoxic in prostate. Under the assumptions underlying the consumption and supply hypoxia model, the lower oxygenation in prostate cancers can be explained by the lower ADC values. Normalizing the data by the mean results in a CSH model that ignores the inherent difference in distribution of ADC and $f_{p}$ in different tissues. Such a model would implicitly assume that similar relationships between ADC and $f_{p}$ gives rise to hypoxia in both prostate and breast tumors. Ideally, we want a method of scaling the data in such a way that the magnitude of ADC and $f_{p}$ is comparable without negating the inherent difference between the tissues. This can be achieved by re-scaling the data using a variant of min-max normalization, where the normalization and data centring is done using global, tissue independent, values $ADC_{low}$, $ADC_{high}$, $f_{p,low}$, and $f_{p,high}$: (4)$$\begin{array}{c} \bar{ADC}=\frac{ADC-ADC_{low}}{ADC_{high}-ADC_{low}}\\ \bar{f_{p}}=\frac{f_{p}-f_{p,low}}{f_{p,high}-f_{p,low}} \end{array}$$
where $\bar{ADC}$ and $\bar{f_{p}}$ are the re-scaled feature arrays. We have assumed that ADC values above $2\times 10^{-3}$ mm${}^{2}$/s represents necrotic tissue, and that ADC values below $0.4\times 10^{-3}$ mm${}^{2}$/s represents the maximum extracellular diffusion restriction possible, where intracellular water diffusion is present. We also assume that the clinically relevant range of the $f_{p}$ values is 0–0.4. The values used for re-scaling the data are these assumed physiological limits of ADC and $f_{p}$ respectively: $ADC_{low}=0.4\times 10^{-3}$ mm${}^{2}$/s, $ADC_{high}=2.0\times 10^{-3}$ mm${}^{2}$/s, $f_{p,low}=0$ and $f_{p,high}=0.4$.

Rather than using a decision boundary to create a binary classifier, we calculated a continuous hypoxia score that reflects the individual weighing of $\bar{ADC}$ and $\bar{f_{p}}$. This score is based on the Euclidean distance from the origin in a cartesian plot of $\bar{ADC}$ versus $\bar{f_{p}}$. For ease of interpretation, the score is given as: (5)$$HS_{Euclid}=1-\frac{1}{\sqrt{2}}\sqrt{ADC^{2}+f_{p}^{2}}$$

This ensures a score ranging from 0 to 1, that increases with increasing estimated probability of hypoxia.

### 2.5. Validation Cohort

To validate the pan-cancer feasibility of any proposed generalized applications developed in the breast tumors, a validation cohort consisting of 95 prostate cancer patients was used. This is the cohort in which the CSH method was first developed and is well suited to use as a validation cohort, as the hypoxia reference standard consists of whole-mount prostatectomy specimens stained with the hypoxia marker pimonidazole. This is considered the gold standard for assessing hypoxia in pathological specimens and provides high resolution hypoxia information that accounts for intra-tumor heterogeneity. Based on the extent and intensity of pimonidazole staining, tumors were categorized as hypoxic or non-hypoxic. Details on the methodology concerning the pimonidazole staining, and hypoxia scoring is available in [19]. Details on the patient and tumor characteristics, as well as the details on the MRI protocol are available in [7].

### 2.6. Statistical Analysis

Non-Gaussian distributed data groups were compared using the non-parametric Mann-Whitney U test and Spearman’s rank correlation coefficient. In the comparison between hypoxia scores, hypoxic fractions and molecular hypoxia scores, a one-sided Mann-Whitney test was used. For all other comparisons, the test was two-sided. Categorical variables were compared using the two-sided Fisher’s exact test. Significance level was set at p<0.05.

## 3. Results

### 3.1. Molecular Hypoxia Score

Tumor hypoxia was assessed in pre-treatment diagnostic tumor biopsies from mRNA expression data by calculating the molecular Buffa hypoxia score ($HS_{mol}$) in 69 breast cancer patients. Large differences in $HS_{mol}$ were found between patients, with a median value of 0.007 and a range from least hypoxic, −0.6 to most hypoxic, 1.4. Patients were categorized as more hypoxic (n = 35) and less hypoxic (n = 34) using the median value as a cut-off. There were no significant differences in clinical parameters like tumor volume, clinical stage or lymph node status between the two hypoxia groups There was, however, an increased occurrence of hypoxic tumors in the invasive lobular carcinomas (p=0.02), and decreased occurrence of hypoxic tumors in estrogen receptor negative tumors (p=0.04) (Table 1). The hypoxia categories served as a hypoxia reference standard for developing a hypoxia imaging tool from DW MR images.

### 3.2. Individual IVIM Parameters and Buffa Hypoxia Score

ADC and $f_{p}$ images were produced, and tumor voxels extracted from 69 breast cancer patients. Both ADC and $f_{p}$ showed significant heterogeneity both between and within patients (Figure 2), demonstrating the ability of imaging to reflect the heterogenic cell growth and vascular pattern characteristic of breast cancer.

Both ADC and $f_{p}$ were found to be significantly lower in the more hypoxic tumors than in the less hypoxic tumors (p<0.02, p<0.05). These results formed the rationale for using ADC and $f_{p}$ as proxies for oxygen consumption and supply, respectively, and combine them using the CSH model to create hypoxia images.

### 3.3. CSH Imaging in Breast Cancer

CSH calculations in breast cancer were first done by directly transferring the linear decision boundary for hypoxia classification found in prostate cancer, as described by Equation (Equation 2), with $ADC_{0}=0.79\times 10^{-3}$ mm${}^{2}$/s and $f_{p,0}=0.437$. This approach gave relatively low hypoxic fractions, mean = 0.1, with limited variation between patients, and there was no difference in estimated hypoxic fractions between the two hypoxia categories (p=0.1). Hence the CSH model cannot be directly transferred from prostate cancer to breast cancer. Comparing the ADC and $f_{p}$ from the prostate cancer patients, on which the original model was trained, to those in the breast cancer patients investigated here may explain these results. Figure 3 shows the distribution of ADC and $f_{p}$ values of all tumor voxels in the breast cancer and the prostate cancer patients.

The median ADC tumor values in the breast cancer patients were 52% higher than those in the prostate cancer tumors, while $f_{p}$ values were similar between the two diseases. This may explain why lower hypoxic fractions were found in breast cancer than what was reported using this method in prostate cancers.

It is clear that a combination of ADC and $f_{p}$, trained and optimized in prostate cancer to identify hypoxia, do not identify hypoxia in breast cancer. If a CSH model capable of identifying hypoxia in breast exists, a retraining for the new cancer type is required. Using the hypoxia reference ($HS_{mol}$) as ground truth, the potential results obtained from model retraining were examined. Hypoxic fractions were calculated using a range of decision boundaries, obtained from different combinations of $ADC_{0}$ and $f_{p,0}$. For each decision boundary, the Mann-Whitney *p*-value was calculated, and used as a measure of the ability of the model to separate between hypoxic and non-hypoxic tumors. *p*-values were calculated for $ADC_{0}$ ranging from 0.4 to 2, and $f_{p}$ ranging from 0.1 to 0.6. Figure 4 illustrates how two different decision boundaries look in one single patient (A) and in the entire patient cohort (B).

The Mann-Whitney *p*-values for all combinations of $ADC_{0}$ and $f_{p,0}$ are shown in Figure 4C. There exists a range of $ADC_{0}$ and $f_{p,0}$ that gives decision boundaries with an ability to separate hypoxic and non-hypoxic tumors that is better than both ADC and $f_{p}$ alone (p<0.005). The best separation (p=0.002) was found for $ADC_{0}=1.51$ mm${}^{2}$/s, and $f_{p,0}=0.33$.

The results above demonstrate that CSH imaging has the potential to provide information on tumor hypoxia in breast cancer, however, it also highlights the fact that different tumor entities require retraining the model against a hypoxia reference standard. A hypoxia reference standard is not commonly available, and a general approach for transferring the CSH model between cancer types, without the need for retraining the model, would expand the applicability of CSH imaging considerably. The development of a pan cancer CSH model approach is described in the following results. As a means of transferring model-parameters between cancer types, the $ADC_{0}$ and $f_{p,0}$ for breast cancer were scaled according to the distributions of ADC and $f_{p}$ in breast cancer, relative to that reported for prostate cancer, according to Equation (Equation 3). Figure 3 shows that the distributions of ADC and $f_{p}$ values are different in breast and prostate. Particularly ADC is higher in breast, whereas the difference in $f_{p}$ between the two cancer entities is much smaller. A model optimized in prostate cancer would therefore tend to overestimate the importance of the relative weighing of the consumption term in tissues with higher ADC values. To obtain a model that preserves the relative weighing of the consumption and supply terms, the model coefficients were re-scaled such that the relationship between them and the mean values of their respective consumption and supply parameters were equal in both tissues. The new, tissue specific model coefficients were calculated according to Equation (Equation 3).

The new, tissue-specific, recalibrated values ($ADC_{0}^{breast}=1.18$ mm${}^{2}$/s, and $f_{p,0}^{breast}=0.43$), gave a decision boundary that resulted in hypoxic fractions that were higher than those obtained using the prostate-trained parameters (mean = 0.32±0.17), with a larger variation between patients. Using this approach, a moderate improvement in separating more hypoxic from less hypoxic tumors was achieved (p<0.02). This improvement was, however, not enough to approach the optimal results identifiable by a retrained model, indicating that the optimal weighting of the two model parameters had not been identified by this model parameter re-scaling.

The generalized CSH method allows creation of continuous hypoxia maps of the whole tumor with the same spatial resolution as the DW MR images. Figure 5 shows the calculated hypoxia maps for two of the patients with low and high $HS_{mol}$, respectively.

Median $HS_{Euclid}4$ was significantly correlated to $HS_{mol}$ (Spearman ρ=0.31, p<0.01). In the binary hypoxia categories, median $HS_{Euclid}$ was significantly higher in the more hypoxic tumors(median $HS_{Euclid}=0.39$) than in the less hypoxic tumors (median $HS_{Euclid}=0.33$), p=0.001. This significance level was similar to the level obtained from the optimal linear decision boundary identified in Figure 4. Figure 6 shows how the three different methods of identifying tumor hypoxia using the CSH-concept compare.

The stratification of tumors into more and less hypoxic was done by dividing the tumors into two equally populated groups, based on the Buffa gene expression score, $HS_{mol}$. To investigate whether the results were influenced by this arbitrary division of hypoxic and non-hypoxic tumors, a broad sweep of $HS_{mol}$ cut-off values was performed. As shown in Figure 7 the generalized CSH method is reliable (p<0.05) for a wide range of cut-off values.

The generalized CSH method and the linear decision boundary CSH method were compared by calculating hypoxic fractions from $HS_{Euclid}$, as the fraction of tumor voxels that satisfies $HS_{Euclid}>HS_{threshold}$, where $HS_{threshold}$ is defined as the median value of $HS_{Euclid}$ across all voxels in all tumors ($HS_{threshold}=0.54$).

The difference between the hypoxic fractions calculated using $HS_{Euclid}$ and the hypoxic fractions calculated using the linear decision boundary approach, were calculated for a range of decision boundaries, given by different combinations of $ADC_{0}$ and $f_{p,0}$. For each decision boundary, the difference in calculated hypoxic fractions was given as $\Delta HF=HF_{LDB}-HF_{Euclid}$ and is shown in Figure 8.

The two calculations of hypoxic fractions were equal, to within 1%, in a region that lies within the region previously identified as giving the best linear decision boundary results (p<0.005).

### 3.4. Validation in a Prostate Cohort

The unsupervised way of calculating hypoxia, using the continuous hypoxia probability score performed well in breast cancer, and reproduced the best achievable results when using a trained linear decision boundary method. The potential of the generalized CSH model was evaluated using the prostate cancer patient cohort from the original paper introducing the CSH methodology [7]. $HS_{Euclid}$ was calculated in the prostate cancer cohort, and hypoxic fractions were calculated using the median $HS_{Euclid}$ as a threshold. There was strong correlation between hypoxic fractions, calculated from the two different methods (Pearson r=0.78, $p<10^{-19}$). The unsupervised, generalized CSH method resulted in an equally good separation between the more hypoxic and less hypoxic tumors in the validation cohort as the linear decision boundary. Figure 9 shows how the generalized method compares to the original classification by Hompland et al. [7].

## 4. Discussion

Most solid tumors contain regions of low oxygenation or hypoxia. Tumor hypoxia has been associated with a poor clinical outcome and plays a critical role in chemo- and radio- resistance. This is the first study to show that consumption and supply imaging, based on DW MRI, depicts hypoxia in breast cancer. Using the breast cancer data, we developed a generalized method for CSH imaging that, as opposed to previous methods, requires no model training. The generalized method was validated, without adaptation, in a prostate cancer cohort.

Our findings in breast cancer follows previous results in prostate [7,20] and cervical [8] cancer. Hompland et al. compared CSH derived hypoxic fractions to hypoxic fractions from pimonidazole-stained whole-mount prostatectomy specimens in 114 patients. They found a strong correlation between the hypoxic fractions, and a strong correlation to disease aggressiveness. Chen et al. used the Hompland-model on 75 prostate cancer patients and found that the hypoxic fraction separated low- and high-grade prostate cancers. Hillestad et al. studied xenografts in mice, and cervical cancer patients. They found strong association between CSH derived hypoxia level, direct measures of hypoxia from pimonidazole staining, and indirect measures by expression of nine hypoxia-associated genes. All three studies relied on a supervised learning procedure to optimize the combination of consumption and supply parameters against a hypoxia reference standard. The generalized method proposed here requires no model training. We found that the generalized method performed similarly to the pre-trained linear decision boundary model [7] in both breast and prostate cancer patients.

The CSH methodology assumes that cellular density reflects oxygen consumption, and that perfusion fraction reflects oxygen supply. Inherently, these assumptions ignore differences in cellular respiration, and differences in blood oxygen saturation and vessel perfusion. It is known that proliferating cell consumes more oxygen than non-proliferating cells [21]. The strong correlation between cell density and the cellular proliferation marker Ki67 [22] support the assumption that oxygen consumption increases with increasing cell density. Treatment that affects tumor cell metabolism may alter the validity of this assumption. A prerequisite for oxygen supply is the presence of functional vessels. It is generally accepted that there is a linear relationship between the blood volume and the IVIM perfusion fraction [16,23,24] and the correlation between the $f_{p}$ and microvessel density was also confirmed in the prostate cancer cohort [7]. In this paper we have used only blood volume to model oxygen supply and therefore assumed uniform transport of oxygen in the blood. However, other factors such as blood rheology, haematocrit and oxygen-haemoglobin saturation also influence oxygen supply, and could alter the relationship between blood volume and oxygen supply. Comparing $f_{p}$ to more oxygen-specific imaging techniques, such as T2* BOLD imaging as proposed by Zhang et al. [25] and Hoskin et al. [26] might provide insight into the nature of the relationship between $f_{p}$ and oxygen supply, potentially improving the precision of the CSH method.

In the breast cancer cohort, the molecular hypoxia score, $HS_{mol}$ was used as the hypoxia reference standard. The correlation between CSH-derived hypoxia and $HS_{mol}$ in the breast cancer cohort was not as strong as that obtained between the CSH-derived hypoxia and pimonidazole-score in the prostate cancer cohort (Figure 9). A possible explanation could be that $HS_{mol}$ was obtained from one or two needle biopsies, originating from unknown locations within the tumors. Given the heterogenic nature of the tumor microenvironment, including vascular and cellular density, the gene expression hypoxia might only be moderately representative of the overall hypoxia status of the tumor. In the validation cohort, for which there existed a much more comprehensive reference standard, the results were much stronger, using either method.

We have found a method for combining images reflecting oxygen consumption and supply to depict hypoxia levels. Hypoxic fraction has been shown to be superior to median hypoxia levels in predicting clinical outcome [12]. However, the hypoxia level defining the hypoxic fractions, relevant for clinical outcome, has been shown to be different in different cancers, and for different treatment regimes [12]. The original CSH method is, by nature, a binary classifier that returns hypoxic fractions, based on an optimization against a pathologic measure of hypoxic fractions. Using the generalized method, the result is a continuous CSH score, where a threshold must be determined that defines the binary classification of hypoxic or non-hypoxic. In this study, the median CSH probability score was used as a threshold. This gave strong likeness to the hypoxic fractions as calculated using the optimal linear decision boundary in both breast and prostate cancers. It is, however, not known how this threshold affects how the hypoxic fractions are related to clinical outcome.

Tumor hypoxia is a spatially and temporally heterogenous phenomenon [2], and reliable longitudinal imaging of oxygenation status is required for therapies aimed at targeting areas of hypoxia in an effort to improve therapeutic outcome [27]. Our proposed model for in vivo hypoxia imaging has several advantages compared to other MRI approaches. It is based on an imaging method already in widespread clinical use, and, as opposed to other proposed techniques, such as OE-MRI, T2* maps from BOLD imaging, requires no image co-registration, contrast agent administration, or oxygen challenge [25,26,28]. Diffusion has high SNR and has been shown to be very stable [29]. CSH imaging is an appealing candidate for dose painting [30] to achieve dose escalation within hypoxic areas because it is non-invasive, the imaging time is a few minutes, and neither contrast media nor ionizing radiation are needed. Furthermore, integration of MRI and linear accelerators is anticipated to revolutionize cancer treatment, allowing real time adaptation of radiotherapy to MR-derived biomarkers of physiological changes, such as hypoxia [31].

The CSH scores were calculated based on ADC and fp estimates obtained from different sets *b*-values. The choice of *b*-values used for calculating the simplified IVIM model parameters may have profound effects on the physiological meaning of the ADC and $f_{p}$ values. In the original paper by Hompland et al. this effect was thoroughly documented, and they found that ADC reflected cellular density when calculated using *b*-values in the range *b* [200,800]. The *b*-values used to estimate ADC and $f_{p}$ in breast were not the same, but within this range, therefore the effect of the selection of *b*-values are unlikely to have significantly affected the estimates of hypoxic fractions.

To what extent our generalized CSH method is transferrable to other cancer entities, with different tissue morphology and underlying biology, remains to be determined. It is, however, worth noting that despite all the similarities between breast and prostate cancers, they have been shown to be different in terms of oxygenation [18]. This is reflected in the findings in this study, where the generalized CSH model gives a higher hypoxia score in prostate cancers than in breast cancers. This difference in hypoxia levels is lost when the tissue specific median $HS_{Euclid}$ is used to calculate hypoxic fractions. Different levels of hypoxia have different biological and clinical implications, demonstrated by the findings of different hypoxia threshold levels for predicting outcome in different cancers [12]. Hillestad et al. found that different levels of hypoxia were associated to biological processes like cancer hallmarks and stabilization of HIF1A protein [8].

## 5. Conclusions

In this study of 68 patients with untreated locally advanced breast cancer we have verified that CSH imaging identifies hypoxic breast cancers. Furthermore, we have developed a novel approach to CHS imaging that requires no model training. The novel unsupervised CSH imaging method for quantification of tumor hypoxia performed equally well as the original CSH model. It also demonstrated a potential for use in multiple cancer entities. The MR technology is widely available and our findings encourages further studies of CSH imaging in other cancer entities and in other setting with the goal being to help overcome hypoxia-induced resistance to treatment.

## Figures and Tables

**Figure 1 cancers-14-01326-f001:**
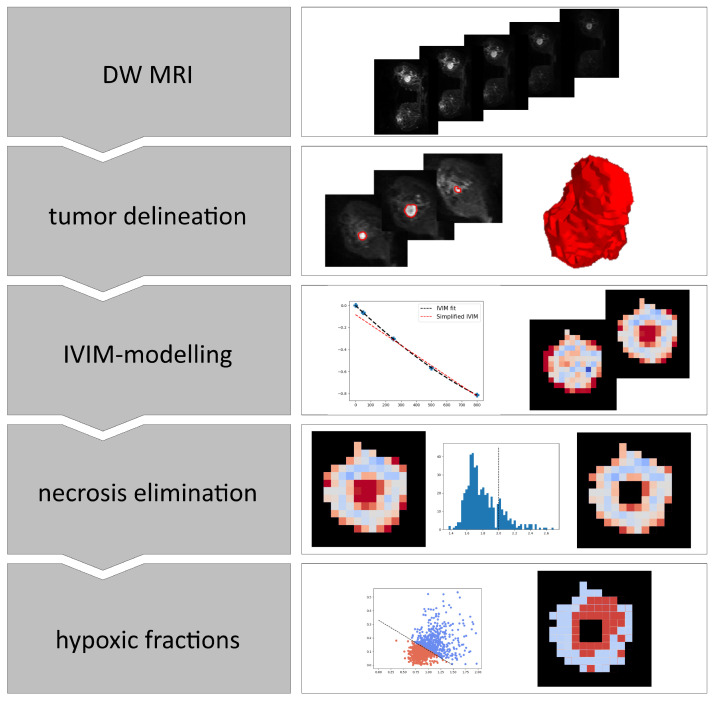
Overview of the steps involved in calculating hypoxia from DWI images. IVIM; Intra voxel incoherent motion.

**Figure 2 cancers-14-01326-f002:**
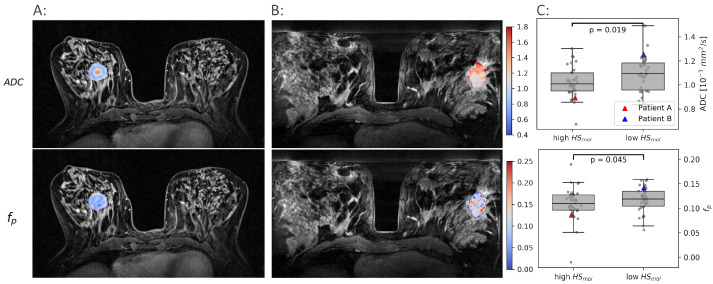
Examples of ADC and $f_{p}$ maps in two patients. One with a high molecular hypoxia score ($HS_{mol}$) (**A**) and one with a low $HS_{mol}$ (**B**). The median ADC and $f_{p}$ for all tumors with a low $HS_{mol}$ are shown against the tumors with a high $HS_{mol}$ in (**C**).

**Figure 3 cancers-14-01326-f003:**
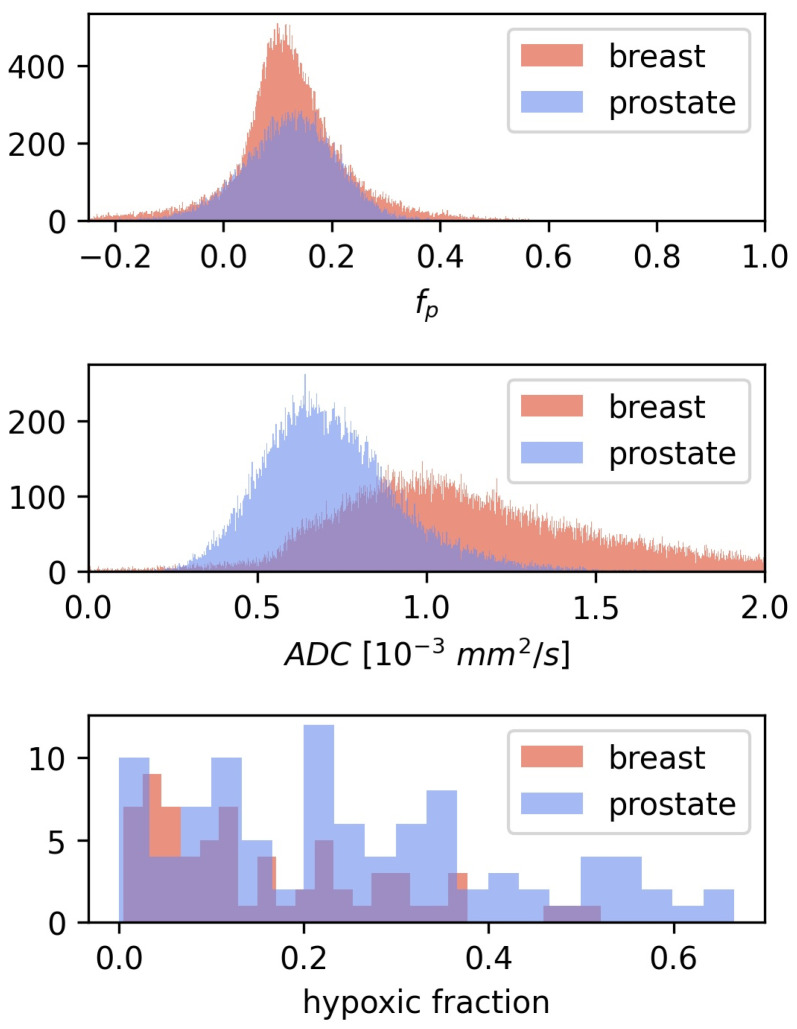
The distribution of ADC and $f_{p}$ for all voxels in the 3D breast tumor volumes, together with the corresponding distributions from the prostate cohort from Hompland et al. [7]. Median ADC and $f_{p}$ in the breast cancer tumors were $1.06\times 10^{-3}$ mm${}^{2}$/s and 0.12 respectively, compared to $0.7\times 10^{-3}$ mm${}^{2}$/s and 0.13 in prostate cancer [7].

**Figure 4 cancers-14-01326-f004:**
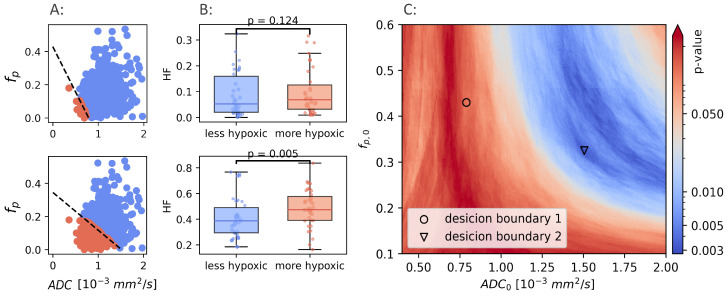
Two different linear decision boundaries (LDB) in a breast cancer patient (**A**). Boundary 1 (upper row) is the same LDB as Hompland et al. found to be optimal in prostate cancer. Boundary 2 (lower row) is the LDB that gives the best results in the breast cancer cohort. The difference in calculated hypoxic fraction between more hypoxic and less hypoxic tumors, using the same two LDBs shown in A (**B**). Mann-Whitney *p*-values calculated using a range of different decision boundaries (**C**).

**Figure 5 cancers-14-01326-f005:**
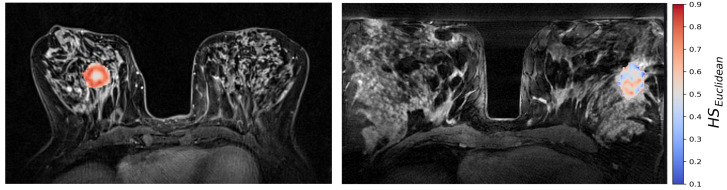
CSH hypoxia maps of the same two patients as shown in Figure 2; one patient with a high molecular hypoxia score $HS_{mol}=0.33$ (**left**), and one with a low molecular hypoxia score $HS_{mol}=-0.23$ (**right**).

**Figure 6 cancers-14-01326-f006:**
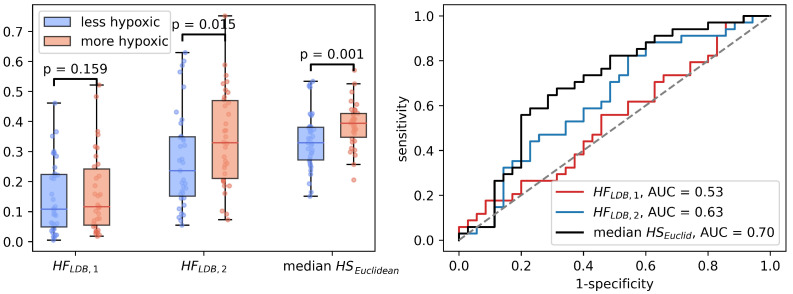
Three different ways of estimating hypoxia in breast cancer using the CSH method. $HF_{LDB,1}$ is hypoxic fraction, calculated using the same model, with the same model-parameters as Hompland et al. used in prostate cancer. $HF_{LDB,2}$ is hypoxic fraction calculated using the Hompland model, but with adapted model parameters, and median $HS_{Euclidean}$ is the median hypoxia score, calculated by Equation (Equation 5). Area under the reciever operateor characteristic curves for all three approaches are shown to the right.

**Figure 7 cancers-14-01326-f007:**
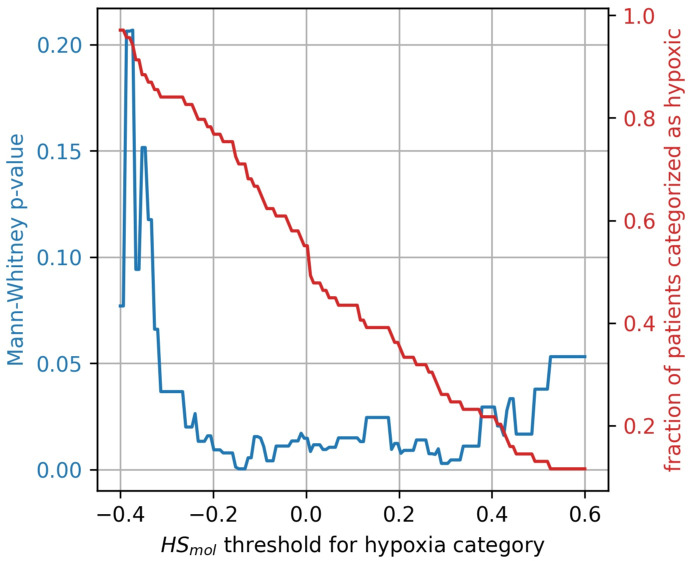
Mann-Whitney *p*-value (blue) for hypoxia stratification thresholds ranging from $HS_{mol}=-0.4$ to $HS_{mol}=0.6$. The resulting fractions of patient stratified as hypoxic for the thresholds are displayed in red.

**Figure 8 cancers-14-01326-f008:**
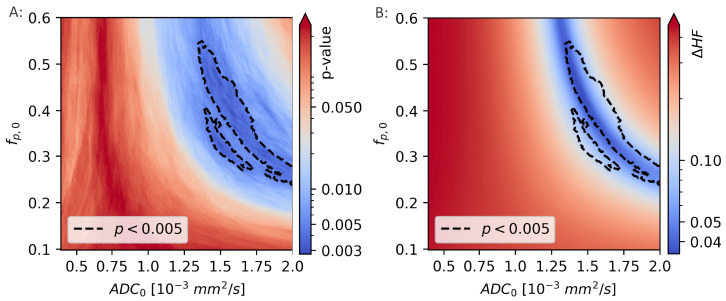
The Mann-Whitney *p*-values from comparisons of the hypoxic fraction (HF), as calculated using a range of linear decision boundaries, with the molecular hypoxia score ($HS_{mol}$) (**A**). The region surrounded by the black, dotted line is indicating the model coefficients ($f_{p,0}$ and $ADC_{0}$) that gives the strongest correlation to $HS_{mol}$ (p<0.005). The difference in HF calculated using the novel, unsupervised method and linear decision boundary models (**B**) is minimal for values of $f_{p,0}$ and $ADC_{0}$ that are largely overlapping those that gives the best decision boundary.

**Figure 9 cancers-14-01326-f009:**
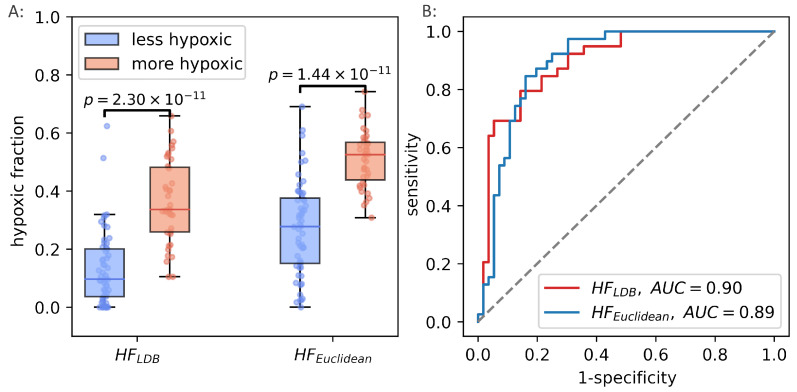
Hypoxic fractions in prostate cancer calculated using the original linear decision boundary($HF_{LDB}$) from Hompland et al., and the hypoxic fraction calculated using the unsupervised (Euclidean) method ($HF_{Euclidean}$) (**A**). Area under the reciever operating characteristic curve for the two methods (**B**).

**Table 1 cancers-14-01326-t001:** Patient and tumor characteristics.

Characteristic	All N (%)	More Hypoxic *	Less Hypoxic *	Adjusted *p*
Patients	69	34	45	
Age (years)				
Mean	49.3	50.3	48.3	1.00 (MW)
Median	49	50	48	
Range	30–70	39–64	30-70	
Clinical tumor stage				0.67 (ANOVA)
T2	21 (30.4)	8 (23.5)	13 (37.1)	
T3	44 (63.8)	25 (73.5)	19 (54.3)	
T4	4 (5.8)	1 (2.9)	3 (8.6)	
Tumor volume (mean cm${}^{3}$)	21.4	22.9	19.9	0.70 (MW)
Lymph node status **				1.00 (ANOVA)
cN0	35 (50.7)	18 (52.9)	17 (48.6)	
cN1	6 (8.7)	2 (5.9)	4 (11.4)	
pN1	28 (40.6)	14 (41.2)	14 (40.0)	
Type				0.02 (Fisher exact)
IDC	55 (79.7)	22 (64.7)	33 (94.3)	
ILC	14 (20.3)	12 (35.3)	2 (5.7)	
Grade				0.12 (ANOVA)
1	5 (7.2)	3 (8.8)	2 (5.7)	
2	50 (72.5)	28 (82.4)	22 (62.9)	
3	13 (18.8)	2 (5.9)	11 (31.4)	
N/A	1 (1.4)	1 (2.9)	0 (0.0)	
ER status				0.63 (ANOVA)
Positive	58 (84.1)	33 (97.1)	25 (71.4)	
Negative	11 (15.9)	1 (2.1)	10 (28.6)	

Abbreviations: IDC = invasive ductal carcinoma, ILC = invasive lobular carcinoma, ER = estrogen receptor, MW = Mann-Whitney, ANOVA = Analysis of variance, N/A = not available. * Hypoxia category is determined by a median split of hypoxia gene expressions as described by Buffa et al. [10]. ** cN1: palpable malignant nodes not verified by fine needle aspiration; pN1: malignant cells in nodes verified by fine needle aspiration.

## Data Availability

The imaging datasets generated and analysed during the current study are available from the corresponding author upon request. The microarray dataset supporting the creation of the molecular hypoxia score is available in the ArrayExpress database, accession number E-MTAB-4439 (http://www.ebi.ac.uk/arrayexpress, (accessed on 17 January 2022)). The trial is registered in the http://www.clinicaltrials.gov/ (accessed on 17 January 2022) website; identifier NCT00773695. Registered on 16 October 2008.

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
