# Peer review of "Quantification of Tumor Hypoxia through Unsupervised Modelling of Consumption and Supply Hypoxia MR Imaging in Breast Cancer"

_cancers, 2022, doi:10.3390/cancers14051326_

Round 1
Reviewer 1 Report
The paper is well written, with statistical tests conducted clearly and results very interesting. I have no issues to raise.
Author Response
We appreciate the positive feedback.
Reviewer 2 Report
The submitted paper aims to investigate if consumption and supply hypoxia (CSH) MRI can depict breast cancer hypoxia, using the CSH-method initially developed for prostate cancer, and to develop a generalized pan-cancer application of the unsupervised CSH-method. The study enrolled 69 patients with breast cancer who underwent biopsy to calculate molecluar hypoxia score from gene expression, and intravoxel incoherent motion (IVIM) to obtain hypoxic fraction based on perfusion fraction (fp) and apparent diffusion coefficient (ADC) in each voxel. In addition, the study separated
patients into two groups (low and high hypoxia) based on the molecular hypoxic score, and utilized the mothed proposed by Hompland et. al., to investigate the capability of MRI in depicting the extent of hypoxia within breast tumors. Authors proposed an unsuperivsed approach to separate low hypoxia from high hypoxia.
Major issues:
1) Line 123. The b-values (0, 50, 250, 500, 800) used to acquire DWI data of breast cancer were different from those for prostate cancer in previous study by Hompland et.al.. In MRI, different b-value settings are known to affect the IVIM results, so as hypoxic fraction.
2) Lines 150-154, b50 was discarded from the analysis in this study. In IVIM theory, low b-value image is important for estimating fp, so neglecting b=50 seems suboptimal to obtain the IVIM parameters (fp and ADC). Although authors stated that "a more stable result can be obtained by estimating fp and ADC only from the signal at b>200 s/mm2", I still have a concern about this issue. Did author perform image smoothing to reduce raw noise before fitting IVIM parameters?
3) Line 250. The development of a pan-cancer CSH model should be described in Method section. The description for the unsupervised method in the Result section is suboptimal.
Minor issues:
1) Line 7. Delete one word "study".
2) Line 50. Abbreviation of MRI should be firstly defined here.
3) Line 102. RIN should be spelled out.
4) Line 171. Did the CSH method for prostate cancer use the same b-value settings as for breast cancer?
5) Line 294. I would like to know whether these definitions (ADC_low, ADC_high,fp_low, and fp_high) were optimized? To what extent the results were affected?
6) Figure 8. (A) and (B) should be indicated in the figure.
7) Figure 9. "desicion" should be corrected to "decision".
Round 2
Reviewer 2 Report
The revision is adequate for publication after the following correction.
In Figure 09. (A) and (B) should be also indicated in the figure.